# Nano-Based Approved Pharmaceuticals for Cancer Treatment: Present and Future Challenges

**DOI:** 10.3390/biom12060784

**Published:** 2022-06-04

**Authors:** Francisco Rodríguez, Pablo Caruana, Noa De la Fuente, Pía Español, María Gámez, Josep Balart, Elisa Llurba, Ramón Rovira, Raúl Ruiz, Cristina Martín-Lorente, José Luis Corchero, María Virtudes Céspedes

**Affiliations:** 1Grup d’Oncologia Ginecològica i Peritoneal, Institut d’Investigacions Biomédiques Sant Pau, Hospital de la Santa Creu i Sant Pau, 08041 Barcelona, Spain; frodriguezl@santpau.cat (F.R.); pcaruana@santpau.cat (P.C.); raulruizpalacios17@gmail.com (R.R.); 2Servicio de Cirugía General y del Aparato Digestivo, Hospital HM Rosaleda, 15701 Santiago de Compostela, Spain; ndelafuente@hmhospitales.com; 3Department of Obstetrics and Gynecology, Hospital de la Santa Creu i Sant Pau, Universitat Autònoma de Barcelona, 08041 Barcelona, Spain; mespanoll@santpau.cat (P.E.); ellurba@santpau.cat (E.L.); rroviran@santpau.cat (R.R.); 4Department of Pharmacy, Hospital de la Santa Creu i Sant Pau, 08041 Barcelona, Spain; mgamezl@santpau.cat; 5Department of Radiation Oncology, Hospital de la Santa Creu i Sant Pau, 08041 Barcelona, Spain; jbalart@santpau.cat; 6Department of Medical Oncology, Hospital de la Santa Creu i Sant Pau, 08041 Barcelona, Spain; cmartinl@santpau.cat; 7Institut de Biotecnologia i de Biomedicina and CIBER-BBN, Universitat Autònoma de Barcelona, Bellaterra, 08193 Barcelona, Spain

**Keywords:** nanomedicine, cancer therapy, nanotechnology, approved nanopharmaceuticals, targeted therapy

## Abstract

Cancer is one of the main causes of death worldwide. To date, and despite the advances in conventional treatment options, therapy in cancer is still far from optimal due to the non-specific systemic biodistribution of antitumor agents. The inadequate drug concentrations at the tumor site led to an increased incidence of multiple drug resistance and the appearance of many severe undesirable side effects. Nanotechnology, through the development of nanoscale-based pharmaceuticals, has emerged to provide new and innovative drugs to overcome these limitations. In this review, we provide an overview of the approved nanomedicine for cancer treatment and the rationale behind their designs and applications. We also highlight the new approaches that are currently under investigation and the perspectives and challenges for nanopharmaceuticals, focusing on the tumor microenvironment and tumor disseminate cells as the most attractive and effective strategies for cancer treatments.

## 1. Introduction

More than ten million people are diagnosed with cancer annually [1]. Cancer comprises a wide diversity of diseases, all of them characterized by numerous cellular physiological systems leading to abnormal and non-stop cell growth in a specific tissue location, forming the tumor [2].

After cancer diagnosis, combined therapies are commonly used, applying different modalities such as surgery followed by chemotherapy and/or radiotherapy. Currently, and despite the advances in conventional treatment options, cancer therapy is still far from optimal. Conventional chemotherapeutic drugs are non-selective small molecules that distribute all over the body without discriminating healthy from damaged tissues [3]. This lack of specificity is translated into the appearance of many severe undesirable side effects and multiple drug resistances [4]. Nanomedicine offers a versatile platform of biocompatible and biodegradable drug-delivery systems (DDS) that can deliver conventional chemotherapeutic drugs in vivo, increasing their bioavailability and concentration around tumor tissues, improving their pharmacokinetics and release profile, and minimizing side effects [5]. The lipid-based and protein-based DDS are the majority of nanopharmaceuticals approved by the U.S. Food and Drug Administration (FDA) and the European Medicines Agency (EMA) for cancer treatment. Innovations in liposome technology, and the incorporation of micelles, polymeric nanomaterials, and inorganic-based nanoparticles, together with the application of targeting with a plethora of ligands, have provided a new generation of nanopharmaceuticals under evaluation in clinical trials, alone or in combination with conventional treatments [6].

Importantly, the recent discoveries of new molecular and cellular targets involved in cancer progression and metastasis are emerging as key factors in nanopharmaceutical development and underscore the importance of continuous feedback between basic and clinical fields. It will choose the best options to overcome the biomedical and technical challenges and will provide the rationale for developing more effective nanopharmaceuticals for cancer therapy. In addition, nanopharmaceutical production, costs, and toxic effects limit the use of nanoparticles in clinical practice. Their toxicity profile requires further extensive research, being one of the most relevant issues required for the eventual approval of nanopharmaceuticals.

In this review, we will provide an overview of the approved and marketed nanopharmaceuticals for cancer treatment and the rationale behind their design and application. Moreover, we highlight the new approaches in DDS development and the nanopharmaceuticals that are currently under clinical trials and discuss their advantages, limitations, and future perspectives.

## 2. Rationale of Nanopharmaceutical Development for Cancer Therapy

Many antitumor agents are available in the market and have been extensively used in clinic. However, these conventional therapies are mainly based on systemically administered non-selective chemotherapeutics of small molecular size (<6 nm) that are below the renal filtration cut-off (around 7 nm). They are excreted through the kidney, which results in a short circulation time and in a wide biodistribution all over the body without discriminating healthy from damaged tissues, especially targeting those tissues that have a high replacement rate such as intestinal lining and immune cells [7]. This lack of specificity is translated into many severe undesirable side effects in patients, such as anemia, appetite loss, constipation, bleeding, diarrhea, fatigue, fertility issues, and hair loss, as the most prevalent signs of toxicity [8]. In addition, low-weight chemotherapeutics have other limitations apart from systemic toxicity, such as low response rates, the development of resistances, and the absence of tumor selectivity. Consequently, all the intrinsic limitations of conventional cancer therapies and the alarmingly high number of deaths in cancer disease have prompted the development of nanomedicine. This field is intensively working on the application of nanotechnology for the development of more effective and safer cancer treatments in the nanoscale range (desirably between 8–100 nm). Since the 1950s, DDS have been explored as alternative engineered platforms for the improvement of cancer chemotherapy. DDS, as nanovehicles, are of several material types, forms, and natures. They offer targeted versions of current anticancer drugs, enhancing antitumor efficacy due to a controlled release at the target site. DDS overcome the non-specific systemic distribution and the inadequate drug concentrations in the tumor site, the intolerable cytotoxicity in healthy tissue, and drug resistance [4,8].

## 3. Targeting Tumor Microenvironment in Cancer Therapy

In the past, tumors were considered as a whole population of cancer cells. The cancer treatment was based on targeting the highly proliferating tumor cells using mainly cytostatic agents designed to target the intrinsic cancer cell mechanisms. However, during cancer progression, tumors become highly heterogeneous, containing different cellular populations characterized by distinct molecular features and different responsivity to therapies. This heterogeneity is provided by many bulk cancer cells, as well as a small population of cancer stem cells (CSCs) and the tumor microenvironment (TME). TME is comprised of the non-cancerous cells and the factors and proteins produced by them, which ultimately support the growth of cancer cells [9]. All of them are critical components, influencing initiation, progression, and metastasis, as well as promoting the resistance of cancer cells to therapies, eventually resulting in patient relapse [10]. This heterogeneity is the main cause that limits the therapeutic value of many drugs [4], and it is further associated with a fatal disease [10,11]. To date, many efforts have been made to understand these multicomponent scenarios and design efficient and precise therapies to target the different compartments in order to increase the clinical impact of cancer treatments [12].

### 3.1. Targeting Cancer Stem Cells in TME

In the last decade, the CSCs hypothesis has gained attention for targeted intervention in cancer therapy. These cancer cells contribute to tumor initiation, growing, resistance, metastasis, and relapse [13,14]. CSCs have been defined as a subset of tumor cells with the ability to self-renew and differentiate into non-CSC highly proliferative cancer cells within the tumor mass [15]. In in vivo experimental model systems CSCs have the ability to initiate tumors and sustain constant tumorigenesis. CSCs share several of their defining features with normal stem cells, including relative quiescence, active DNA repair systems, aggressive proliferation, and drug resistance [16]. These processes are principally regulated by the WNT/β-catenin, transforming growth factor-β, Hedgehog, and Notch signals [14]. Three origins have been postulated for CSCs: (a) (epi)genetic changes such as methylation, demethylation, mutations, and rearrangement in the stem/progenitor cell population (niche) or even in differentiated cells [17]; (b) spontaneous oncogenic reprogramming in somatic cells; and (c) TME activation through providing extracellular cues [18].

CSCs are identified from the expression of several cell surface markers, and these markers are tumor-type dependent and conform to a signature; thus, the different markers are combined to isolate a *bonafide* cell population with high numbers of CSCs. However, the gold standard method to unveil the tumor-initiating capabilities of CSCs is the use of limiting dilution in xenograft animals to form tumors (see detailed reviews of CSCs in [19,20]). To date, several studies have been able to prove the existence of CSCs in tumors, and their tumorigenicity has been demonstrated in several cancers, including brain, liver, lung, colon, breast, ovarian, pancreas, prostate, melanoma, head, neck, and bladder [21,22]. Furthermore, their frequency increases upon tumor progression, being different from one cancer to another [23,24]. Several CSCs markers such as CD44, ALDH1 CD133, and CXCR4 have been identified in different tumors and are targets for therapeutic interventions [24].

CSCs are also able to resist conventional therapies such as chemotherapy and radiotherapy [25]. Such resistance is given by the increased expression of drug transporters (e.g., ATP-binding cassette (ABC) membrane transporters), the maintenance of a slow dividing state (quiescence), and efficient DNA repair mechanisms [26]. Moreover, CSCs have enhanced epithelial to mesenchymal (EMT) properties, enhanced expression and activation of several survival signaling pathways, and increased immune-evasion and DNA-repair mechanisms [25].

Furthermore, the functional properties of CSCs during cancer expansion and the responses to the therapeutic approaches are defined by the TME [27,28]. The plasticity of CSCs through transformation into proliferative cancer cells is possibly under the control of signals from both CSCs and the TME. Cancer-associated fibroblasts are among the most influential cells for promoting both the differentiation of CSCs and the dedifferentiation of non-CSCs to a CSC-like phenotype. Interestingly, CSCs can also alter the cellular microenvironment in favor of cancer progression, influencing stromal cells through the release of extracellular vesicles [29].

The elimination of CSCs, based on their preferential expression of markers and/or their supporting elements, is thought to be better in efficiently antitumor therapy. CSC targeting has allowed for the creation of novel technologies and therapies that can target tumor progression at earlier stages of pathogenesis [30,31,32,33]. To date, no approved nanopharmaceutical is used to eliminate these populations, but some of them are in preclinical and clinical trials [34].

### 3.2. Targeting Tumor Stroma in TME

Tumors are complex systems that comprise malignant cancerous cells and the so-called tumor stroma formed by the non-cancerous cells such as fibroblasts, immune cells, bone marrow-derived inflammatory cells, lymphocytes, and cells that form blood vessels, as well as the extracellular matrix (ECM). They also include the proteins produced by all the cells that also support the growth of the malignant cells [35]. The microenvironment of the tumor is an integral part of its physiology, structure, and functioning, and it is an essential aspect of the malignant process [36]. The interactions between the malignant and normal cells create the TME and play critical roles in cancer development, progression, and metastasis. The combination of cancer cell mutations (and other alterations) coupled with changes to the tumor stroma contributes to the tumor heterogeneity and drives tumorigenesis, resulting in fatal disease [37]. Over the years, most anticancer therapies have targeted malignant cancer cells specifically, while largely ignoring TME. Investigations are now focusing and are extensively exploited on the tumor microenvironment as a separate cancer-associated entity. Anticancer therapies should target both cancer cells and the stromal compartment to be effective and result in improved patient outcomes [36].

The use of experimental cancer models has shown the tumor architecture of TME and the complexity of cellular and noncellular interactions within a tumor [37]. The monitoring of changes in molecular and cellular profiles of the tumor microenvironment could be vital for identifying cell or protein targets in tumor progression, being useful for cancer prevention and therapeutic purposes [38].

#### 3.2.1. Angiogenic Targets

Angiogenesis is the formation of new blood vessels and involves the migration, growth, and differentiation of endothelial cells to supply oxygenation and nutrients to tissues. Angiogenesis is highly activated in tumors [39]. However, the rapid growth of the tumor cells results in leaky and highly disoriented vascular structures and the inability of blood vessels to provide oxygenation and nutrients to tumor cells, leading to oxygen-deficient or hypoxic regions in the tumor. This situation results in a hypoxic tumor core, making it immune to chemotherapy and radiation treatments. The lack of oxygen can also cause glycolytic behavior in cells, inducing greater cell migration in-vivo and in-vitro [39]. Controlling the angiogenesis is a critical step in cancer progression, and the dissection of the molecular interactions of this process by the TME enhances prognoses and facilitates targeted therapy [40].

#### 3.2.2. Drug Chemoresistance Targets

TME is also recognized as a key contributor to drug resistance in cancer treatment. When cancer develops, the stroma undergoes changes to become fibrotic and activated. The extracellular matrix (ECM) becomes denser and more rigid, thanks to new connective fibers, such as tenascin and fibronectin, which cancer cells can invade. Cancer-associated fibroblasts (CAFs), myofibroblasts, and mesenchymal stromal cells (MSCs) change shape and expression profiles and become more proliferative. Senthebane et al. [41] reported that components of TME, including CAFs and the ECM, are major contributors to chemoresistance [41]. In response to chemotherapy or radiation therapy, CAFs and MSCs secrete different growth factors, cytokines, and chemokines that promote cancer cell survival, proliferation, invasion, and metastasis, leading to resistance. Dense fibrosis causes limited access of cancer cells to therapeutic agents in three ways: creating an extracellular matrix (ECM) barrier that such agents cannot diffuse through; promoting the stromal cytochrome P450 (CYP)-mediated degradation of drugs; and increasing interstitial pressure that prevents therapeutic agents from entering the tumor [37,41].

Extensive work has been done to explore the interactions between cancer cells and the TME. The monitoring of the changes in the TME using their molecular and cellular profiles could be pivotal for identifying cells or protein targets for cancer prevention and therapeutic purposes. Discovery studies involving the laser capture microdissection of stroma [42], coupled with analyses of genetic mutations [43] and epigenetic changes [44], can uncover novel therapeutic targets. Drugs that target stromal components are in various stages of preclinical and clinical development. In most cases, the stroma-targeting drugs abrogate resistance to cancer cell-targeting therapies, leading to strong anticancer responses [45]. However, some future challenges must be considered: the unknown elements of the optimal composition of combinations comprising stromal-targeting and cancer-targeting agents or even the administration sequence should be defined for being offered to patients [45]. The order of drug administration is important because certain sequences can result in adverse effects. For example, when chemotherapy is administered after the initiation of an immune response, it could cause the death of immune cells, thereby reversing their cytotoxic effect. As we continue to better understanding the complex interactions between the heterogeneous milieu of cellular and non-cellular contributors to the TME, we will be able to improve stroma-targeting strategies and design more effective anticancer therapies.

### 3.3. Disseminated Tumor Cell Targets

One of the major hallmarks of cancer is the dissemination of tumor cells to adjacent organs or to distal sites (or metastasis). Metastasis comprises approximately 90% of total cancer-related deaths [46]. Conventional cytostatic agents are successful at reducing primary tumor size, but they have a poor effect on disseminated tumor cells. Several studies have described that metastasis-initiating cells (MICs) seem to be responsible to tumor growth in distant organs. MICs result from (i) a phenotype with intrinsic programs to survive the stresses of the metastatic process, (ii) undergo epithelial–mesenchymal transitions, (iii) enter slow-cycling states for dormancy, (iv) escape immune surveillance in collaboration with the metastatic microenvironment, and (v) establish supportive interactions with organ-specific niches [47,48,49].

Thus, one of the most promising strategies to control cancer is focused on designing drugs that interfere with MICs (see detailed review in [48]). MICs overexpress the chemokine receptor CXCR4. The overexpression of this receptor in primary tumors has been described in 20 different cancer types and contributes to tumor growth, angiogenesis, metastasis, and therapy resistance. CXCR4-overexpressing tumors are likely to metastasize in an organ-specific and CXCL12-dependent manner, with the lung, liver, brain, kidney, skin, and bone marrow being the CXCL12-expressing organs [50,51]. Supporting this fact, the inhibition of the CXCL12/CXCR4 axis resulted in a reduced metastatic load in many cancer mouse models [52,53,54]. Metastatic CSCs overexpressing CXCR4 are clinically relevant targets; thus, their selective elimination could represent an advance in metastases control. A multitude of CXCR4-directed antagonists have been developed during the last decade. Among them, AMD3100 (Plerixafor), a low-weight molecule, is the unique compound approved by the FDA for the mobilization of hematopoietic stem cells and for the treatment of multiple myeloma and non-Hodgkin’s lymphoma [55]. Many other CXCR4 antagonists are being developed, and some of them are under clinical trials. No approved nanopharmaceutical is used to treat cancer stem cells.

## 4. Modes of Targeting: From Size to Multicomplexity DDSs

Because of the nanoscale size, nanovehicles tend to accumulate more in tumor tissues with respect to low-weight drugs due to the enhanced permeability and retention (EPR) effect [56]. This inherent targeting (referred to Passive targeting) is due to the tumoral leaky vasculature, different pH, and different local temperature, as well as a deficient lymphatic drainage system in tumors [57]. However, this passive mode of targeting can induce (a) multi-drug resistance (MDR) due to the high systemic dosage used to achieve a sufficient amount of drug at the tumor site to trigger a therapeutic response, (b) poor drug diffusion, and (c) accumulation in the liver and spleen [58].

Targeted drug-delivery systems (TDDS) are commonly used to overcome the limitations of low specificity, low therapeutic index, low absorption, short half-life, and a large volume of distribution by low-molecular-weight drugs and untargeted nanovehicles. Moreover, TDDS minimizes the adverse effects, reducing nephrotoxicity, neurotoxicity, and cardiotoxicity by making changes in the inappropriate disposition of the drug and reducing its presence in non-targeted areas [59]. At the same time, TDDS could also maximize the therapeutic efficacy of the drug by preventing its inactivation in circulation, avoiding the degradation of the drug before reaching the target site [60,61,62,63].

The functional targeted of nanovehicles can be achieved through two different strategies: (i) active targeting and (ii) stimuli-responsive targeting [64]. Active targeting configures the nanovehicle to target-specific cells after extravasation using the molecular recognition (ligand-receptor or antigen-antibody interactions) of the tumor cells through specific receptors that are overexpressed on them [65]. The ligand must have a high affinity for its receptor, and trigger receptor-mediated endocytosis, after which the intracellular release can be favored by acidic pH or enzyme [66]. These ligands can be either adsorbed onto the particle surface or covalently bound to one of the components of the nanoparticles, normally poly-(ethylene glycol). Different types of ligands can be used to target cancer cells, including, most commonly, folic acid, and peptides or protein-targeting moieties, such as antibodies, anti-body fragments, cell-penetrating peptides (CPPs), growth factors, or cytokines. This interaction is highly specific and strong, and antibodies seem to be the most effective ligands. The production of conventional antibodies is difficult and expensive; therefore, specifically, an amino acid sequence called the antigen-binding fragment (Fab), which univocally binds to antigens, is preferred because of it is safe against non-specific binding and can be easily engineered [67]. Finally, the small synthetic single-stranded RNA or DNA oligonucleotides (normally composed of 20–60 nucleotides), called aptamers, which can form specific shapes (helices or single-stranded loops), are a new type of ligands that are extremely versatile and can bind different kinds of targets—proteins, inorganic molecules, and cells with high selectivity. Their preparation is much simpler and cheaper than the antibodies, not showing any sign of toxicity [68,69].

On the other hand, stimuli-responsive targeting is intended as the localized release of drugs induced by a trigger that alters the structure of the nanocarrier [70]. This system is highly specific and can be activated “on-demand” at the desired site. Stimuli can be internal, such as variations in pH, redox conditions, and ionic strength, or external, such as temperature, ultrasounds, magnetic fields, and ultraviolet/near-infrared (UV/NIR) radiation [71].

DDS can be designed to be responsive to these stimuli and to achieve enhanced release of their cargo in the precise location [72]. Moreover, external stimuli such as local hyperthermia and UV/NIR light can act in the microenvironment, for example, by enhancing the permeability of blood vessels to favor tissue penetration, or ultrasounds can induce the release of contrast agents at the tumor site, while magnetic fields can locally drive DDS, thus triggering drug release via hyperthermia [73,74].

## 5. Approved Nanopharmaceuticals for Clinical Uses in Cancer Therapy

Figure 1 and Figure 2 and Table 1 list the U.S. Food and Drug Administration (FDA) and the European Medicines Agency (EMA)-approved nanopharmaceuticals for cancer therapy. Among the DDS, the lipid-based platform, such as nanoliposomes, supplies a good number of the total approved nanopharmaceuticals for cancer treatment [75]. To date, 56% of them are lipid-based nanoformulations, and the rest are included as protein-based (38%) and metallic-based nanoformulations (6%).

### 5.1. Lipid-Based Approved Nanopharmaceuticals

Liposomes were the first drug-delivery systems in the market and are still the most used to date [75]. Their flexibility of composition, biocompatibility, biodegradability, and non-immunogenicity are their main advantages compared to other DDSs [76]. Liposomes are artificially prepared single or multilamellar phospholipid vesicles. They are 50–100 nm in size and have anionic, cationic, or neutral charges with a central aqueous phase or core [76,77]. Liposomes and micelles are both made of phospholipids, but they differ in their morphology. Liposomes are mainly used to encapsulate hydrophilic drugs in their aqueous core, but hydrophobic drugs can also be accommodated in the bilayer or chemically attached to the particles [76]. Micelles, instead, have a hydrophobic core that can encapsulate hydrophobic drugs [77]. They are exploited for delivering and reaching a specific target site, thus minimizing biodistribution toxicity. [76]. Conventional liposomes, named the first generation of liposomes, displayed a relatively short blood circulation time and mainly accumulated in the liver and spleen, via the uptake of the mononuclear phagocyte system (MPS), reducing their tumor tissue-specific targeting capacity [78]. The selection of different lipid natures and their combinations has overcome these drawbacks. The sterically stabilization of nanoliposomes with sphingomyelin/choline (SM/CHO) and the polyethylene glycol (PEG)-coated version (pegylated; Stealth) have provided a long circulation time and the ability to extravasate through the ‘leaky’ tumor vasculature by using the passive targeting. In addition, pegylated liposomes are less extensively taken up by cells of the reticuloendothelial system (RES) and have a reduced tendency to leak drugs in circulation. All of these chemical modifications allow for the accumulation of the encapsulated drug in tumor tissue compared to the first generation of liposomes [79].

To date, there are various approved nanopharmaceuticals using lipid-based nanotechnology platforms. Doxil^TM^ (Bridgewater, NJ, USA) was the first FDA-approved in 1995 [80]. Like Doxil^TM^, the liposomal-based nanoformulation Caelyx^TM^ (Kenilworth, NJ, USA) and Myocet^TM^ (Castleford, UK) were also approved by the EMA in 1996 and 2000, respectively. The three nanopharmaceuticals, Doxil^TM,^ Caelyx^TM^ (PEGylated), and Myocet^TM^ (non-PEGylated), encapsulate and stably retain the chemotherapeutic doxorubicin [80]. They were formulated to improve the safety profile of Doxorubicin, which is characterized by high cardiotoxicity [80].

Their reduced cardiotoxicity allows for a larger cumulative dose than that acceptable for free doxorubicin [81,82,83]. Doxorubicin is an anthracycline antibiotic that is generally believed to interact with the DNA by intercalation, inhibiting macromolecular biosynthesis [84]. Doxorubicin stabilizes the topoisomerase II complex after it has broken the DNA chain for replication, preventing the DNA double helix from being resealed and stopping the replication process. Another reported mechanism of doxorubicin is its ability to generate free radicals that induce DNA and cell membrane damage [84]. Doxil^TM^, Caelyx^TM,^ and Myocet^TM^ exhibit a prolonged circulation time and a reduced volume of distribution, thereby improving tumor uptake through the EPR effect and extending effective tumor therapy [78,79].

Doxil^TM^ and Caelyx^TM^ are indicated for the treatment of ovarian cancer, AIDS-related Kaposi’s sarcoma, and in combination with bortezomib, for the treatment of multiple myeloma. Caelyx^TM^ and Myocet^TM^ are also indicated for patients with metastatic breast cancer [78].

DaunoXome^TM^ (Craigavon, UK) is another nanoliposomal preparation that encapsulates Daunorubicin, an anthracycline antibiotic with strong antineoplastic activity. Like Doxorubicin, daunorubicin activity has been attributed mainly to its intercalation between the base pairs of native DNA. It causes DNA damage such as fragmentation and single-strand breaks. There are two limiting factors in the use of anthracyclines as antitumoral agents: chronic or acute cardiotoxicity and spontaneous or acquired resistance. Daunorubicin has a particular affinity for phospholipids, and the development of resistance is linked to some membrane alterations [85]. The nanoformulation helps to protect Daunorubicin from chemical and enzymatic degradation in the systemic circulation, minimizing protein binding and reducing the uptake by normal (non-reticuloendothelial system) tissues, thus increasing the accumulation in tumors by passive targeting [85]. DaunoXome^TM^ is indicated as a first line for advanced HIV-associated Kaposi’s sarcoma [86].

Mepact^TM^ (Cambridge, MA, USA) is the liposomal muramyl tripeptide phosphatidylethanolamine, Mifamurtide, an immunomodulator that activates monocytes and macrophages. Mepact^TM^ is indicated in children and young adults for the treatment of high-grade resectable non-metastatic osteosarcoma after complete surgical resection [87].

Ameluz^TM^ (Leverkusen, Germany) is a gel formulation containing INN-5-aminolaevulinic acid in a nanoemulsion, enhancing its penetration into the epidermis. The substance is metabolized to protoporphyrin IX and activated by a red light, forming a reactive oxygen species and destroying the targeted cells. It is indicated for the treatment of actinic keratosis of mild to moderate intensity on the face and scalp [88].

Marqibo^TM^ (Henderson, NV, USA) is a sphingomyelin- and cholesterol-based nanoparticle formulation of Vincristine. It is an antineoplastic drug with a broad spectrum of activity against hematological malignancies and childhood sarcomas. It induces neurotoxicity and peripheral neuropathy in a dose-dependent manner [89]. The liposomal carrier component facilitates the loading and retention of Vincristine; extravasation; and the slow drug releasing in the tumor microenvironment, and it improves the safety profile of vincristine, reducing its side effects [90]. It is indicated in adults with advanced, relapsed, and refractory Philadelphia chromosome-negative acute lymphoblastic leukemia [91].

Onivyde^TM^ (Cambridge, MA, USA) is Irinotecan (DNA topoisomerase I inhibitor) is encapsulated in a lipid bilayer vesicle, which prolongs the circulation time and increases the delivery of Irinotecan in tumors with compromised vasculature. It is indicated for the treatment of the metastatic adenocarcinoma of the pancreas in patients who have progressed following Gemcitabine-based therapy.

Vyxeos^TM^ (Dublin, Ireland) is a liposomal formulation of a fixed combination of Daunorubicin (inhibitor of DNA polymerase activity) and Cytarabine (a cell cycle phase-specific antineoplastic agent). It exhibits a prolonged plasma half-life and accumulates and persists in high concentration in the bone marrow. Vyxeos^TM^ is indicated as a monotherapy for the treatment of adults with high-risk acute myeloid leukemia [92].

### 5.2. Protein-Based Approved Nanopharmaceuticals

Protein-based nanoparticles also offer many advantages, such as biocompatibility and biodegradability. Moreover, their preparation and encapsulation processes do not require the use of toxic chemicals or organic solvents (see the detailed review in [93]). These nanoparticles can be generated using proteins such as fibroins, albumin, gelatin, gliadin, legumin, 30Kc19, lipoprotein, and ferritin proteins, with fibroin and albumin being the most widely used. As DDS, they can carry genetic materials, anticancer drugs, peptide hormones, growth factors, DNA, and RNA. In addition, protein from various sources can be manufactured into nanoparticles using an easy, cost-effective synthesis process. They are also more stable than other DDS. Moreover, the application of protein nanoparticles has shown great potential in the future [94]. Several protein-based nanoparticles have been approved by the U.S. Food and Drug Administration (FDA) and the European Medicines Agency (EMA) for the treatment of cancer. Figure 2 shows the current protein-based marketed nanoparticles.

Oncaspar^TM^ (Suresnes, France) is the PEGylated formulation of the enzyme asparaginase, which breaks and reduces the blood level of the amino acid asparagine, needed for tumor cells to grow and multiply, and so its reduction in the blood causes the cells to die. Normal cells, by contrast, can produce their own asparagine and are less affected by the treatment. This nanoformulation prolongs half-life and can reduce the risk of allergic reactions in contrast to conventional formulations. Oncaspar^TM^ is indicated for the treatment of acute lymphoblastic leukemia in adults and children [95,96].

Ontak^TM^ (Suresnes, France) is a diphtheria toxin-based recombinant fusion toxin that obtained FDA approval for clinical application in 1999. It is indicated for the treatment of human CD25+ cutaneous T cell lymphoma (CTCL). DT mediates its cytotoxic effect through the inhibition of protein synthesis in target cells, and it interacts with the heparin-binding epidermal growth factor receptor (HBEGFR) inducing receptor-mediated endocytosis. In the cytoplasm, the catalytic domain first binds to nicotinamide dinucleotide (NAD) and then transfers an adenosine diphosphate ribosyl (ADPR) moiety to elongation factor 2. Thus, the EF2 is irreversibly inactivated, resulting in the inhibition of protein synthesis and cell death [97,98]. Unfortunately, Ontak^TM^ immunotoxin was discontinued in 2014 due to issues related to its production process in *Escherichia coli* [99]. However, other approaches attempt to improve the production or enhance the potency of Ontak^TM^-derived immunotoxins. Some T-cell malignancies overexpress IL2R, making it an ideal target for immunotoxins [100]. Recently, the development of monovalent and bivalent human IL-2 fusion toxins targeting human CD25^+^ cells using an advanced, unique diphtheria-toxin-resistant yeast Pichia Pastor-is expression system was reported, showing potent and selective binding affinity to cells expressing high-affinity IL-2R [99]. One of the most promising candidates is E7777, which is currently being tested in Phase II and phase III clinical trials (ClinicalTrials.gov identifiers: NCT02676778 and NCT01871727, respectively) [101].

EligardTM (Milano, Italy) is Leuprorelin, a gonadotrophin-releasing hormone (GnRH) agonist. This nanopharmaceutical has a more enhanced potency and a more prolonged duration of action than natural GnRH. Chronic exposure results in the suppression of LH, FSH, and testosterone. EligardTM is indicated for the treatment of advanced hormone-dependent prostate cancer and breast cancer, as well as other pathologies such as endometriosis, uterine myoma, uterine fibrosis, and precocious puberty [102].

Abraxane^TM^ (New York, NY, USA) and Pazenir^TM^ (San Francisco, CA, USA) are the albumin-bound particles forms of Paclitaxel (nab-paclitaxel), which gained FDA and EMA approvals in 2005 and 2008, respectively. Paclitaxel, better known as Taxol, is the first member of the taxane family to be used in cancer chemotherapy. These compounds exert their cytotoxic effect by arresting mitosis through microtubule stabilization, resulting in cellular apoptosis. Specifically, Paclitaxel blocks cells in the G2/M phase of the cell cycle, and such cells are unable to form a normal mitotic apparatus. It is a broadly accepted option in a variety of solid tumors. Being hydrophobic, paclitaxel requires solvents (Cremophor EL or polysorbate) to enable parenteral administration. Moreover, it was reported that Cremophor EL entraps paclitaxel into circulating micelles, which reduces its availability and delivery into tumors and results in nonlinear kinetics with an absence of a dose response relationship (increasing the dose increases toxicity), which limits the efficacy [103].

However, significant toxicities, such as myelosuppression, allergic reactions, neurotoxicity, and systemic toxicity, have mainly been due to solvents, limiting the effectiveness of paclitaxel-based treatments. Both the absence of solvents and the receptor-mediated delivery result in decreased toxicity and increased antitumor activity [104,105]. The nanopharmaceutical form, nab-Paclitaxel, takes advantage of the increased delivery of albumin to tumors via a receptor-mediated transport called transcytosis. Nab-Paclitaxel binds to gp60, the albumin receptor on endothelial cells, which in turn activates caveolin-1 and the formation of caveolae. Caveolae transports the albumin–paclitaxel conjugate to the extracellular space, including the tumor stroma. There, SPARC (secreted protein, acidic and rich in cysteine) that is selectively secreted by the tumors binds to albumin-bound paclitaxel with the resulting release of paclitaxel in the vicinity of tumor cells [106]. In addition, the internalized nab-paclitaxel exhibits significant immunostimulatory activities to promote the cancer-immunity cycle through the synergistic effect in T cell activation, reversing the immunosuppressive pattern of the TME) and acting synergistically with cytotoxic lymphocytes (CTLs) in the clearance of tumor cells [107]. Abraxane^TM^ and Pazenir^TM^ are indicated in metastatic breast cancer, the metastatic adenocarcinoma of the pancreas, and for the treatment of non-small cell lung cancer in patients; they are not candidates for surgery and/or radiation therapy [108,109].

Kadcyla^TM^, (Ulm, Germany) also named T-DM1 or Ado-Trastuzumab emtansine, was the first antibody-drug conjugate (ADC) approved by the FDA and EMA in 2013. It is a 30 nm protein-based nanocompound formed by Transtuzumab, an anti-HER-2 humanized monoclonal antibody that is bound to DM-1 (emtansine, an antimicrotubular agent that blocks cellular division) [110]. Kadcyla^TM^, as a single therapy, is indicated for the adjuvant treatment of adult patients with HER2-positive early breast cancer who have residual invasive disease [111]. It is commonly used in combination with other antineoplastics for the treatment of HER-2-positive breast tumors. Trastuzumab exerts an antitumor activity and *per se* binds to HER-2 receptors that belong to the family of tyrosine kinase receptors and block the cleavage of the extracellular domain, inhibiting the intracellular signaling cascades (MAPK and PI3K/Akt pathways) and, consequently, suppressing cell proliferation and tumor growth. Furthermore, this binding also promotes HER-2 receptor degradation and activates the antibody-dependent cell-mediated cytotoxicity. It is used in combination with other antineoplastics for the treatment of HER-2-positive breast tumors [111,112].

To date, a total of 11 antibody-drug conjugates apart from Kadcyla™ have been approved by the FDA and EMA. [113]. Table 2 summarizes the antibody-drug conjugate trade name, target, company, approval date, and indications. It includes brentuximab vedotin (Adcetris™), inotuzumab ozogami-cin (Besponsa™), gemtuzumab ozogamicin (Mylotarg™), Moxetumomab pasudotox (Lumoxiti™), polatuzumab vedotin-piiq (Polivy™), Enfortumab vedotin (Padcev™), Sacituzumab govitecan (Trodelvy™), Trastuzumab deruxtecan (Enhertu™), belantamab mafodotin-blmf (Blenrep™), lon-castuximab tesirine-lpyl (ZYNLONTA™), and tisotumab vedotin-tftv (Tivdak™) [114,115,116,117,118,119,120,121,122,123,124].

Apart from these nanopharmaceuticals, some nanocompounds are regionally marketed. Lipusu™ is a non-pegylated liposome formulation containing paclitaxel. It is approved in China (2006), and it is indicated for the treatment of non-small cell lung cancer, and ovarian and breast cancers [125]. Genexol™ (Seoyoon Cheong, Korea) and Nanoxel™ (Seoyoon Cheong, Korea) were approved by South Korean (2007) and Indian agencies. They are formed by polymeric micelles and paclitaxel. They are indicated for the treatment of non-small cell lung cancer and breast cancer (in the case of Genexol™) and metastatic breast cancer, non-small cell lung, and carcinoma Kaposi’s sarcoma (in the case of Nanoxel™) [126,127]. Bevetex™ is a nanoformulation of polymeric-lipidic nanoparticles and paclitaxel, approved in India. It is indicated for the treatment of ovarian, breast, and bladder cancers, and Kaposi’s Sarcoma [128].

### 5.3. Metallic-Based Approved Nanopharmaceuticals

To date, NanoTherm^TM^ (Berlin, Germany) is the only metallic-based for cancer therapy that has gained FDA and EMA approvals [129]. NanoTherm^TM^ formulation is a colloidal suspension of amino silane coated with iron oxide nanoparticles suspended and distributed in 15 nm size particles [129]. Magnetic nanoparticles are introduced either directly into the tumor or into the resection cavity wall. They are subsequently heated by an alternating magnetic field, thus destroying the cancer cells or sensitizing them for concomitant radiotherapy or chemotherapy applications to avoid recurrences. This nanoparticle formulation is indicated for the treatment of primary or recurrent glioblastoma multiforme and prostate cancer [130].

## 6. Nanopharmaceuticals in Clinical Trials for Cancer Therapy

Nanomedicine is constantly expanding. Since the early 2000s, almost 30,000 articles have been published, including original research (source MEDLINE/PubMed) and new nanopharmaceuticals for therapy enter clinical investigation every year. Most of them are anticancer and antimicrobial nanodrugs [131]. Concerning only cancer therapy, as of 2021, 55 clinical trials, including the term “nano”, were listed in phase I/II on ClinicalTrials.gov (accessed on 9 May 2022) [132] (Appendix A). Most of the nanopharmaceuticals are new versions of previously approved low-weight drugs and nanoparticles that have already been proven to be successful, such as liposomes and polymers [133]. The remainder of the investigational nanodrugs demonstrate a trend toward agents using micelles, as well as the introduction of formulations using dendrimers; meanwhile, polymer-based nanopharmaceuticals are less prevalent. Although the majority of FDA-approved nanodrugs relied on passive targeting via the EPR effect, next-generation drugs in clinical trials employ active targeting approaches (Figure 3 and Appendix A) [134]. Another new trend is the change from relatively simple nanoparticles to complex and multicomponent drug-delivery platforms [135]. In this regard, modern approaches to protein engineering, as well as advances in polymer and inorganic chemistry, have also resulted in an expansion of novel nanomaterials. This has allowed the initial goals for nanopharmaceuticals (improved PK, efficacy, and safety) to evolve into system designs that allow for more complex functions, such as controlled release and active targeting [135].

On the other hand, inorganic nanoparticles (INPs) have been extensively utilized for diagnostic purposes. For example, gold nanoparticles (AuNPs) have been widely studied due to their biocompatibility and the ease of controlling their size distribution and shape. Furthermore, the surface chemistry of AuNPs can also be easily modified through conjugation with various polymers, antibodies, small-molecule therapeutics, and molecular probes. Another prevalent type of INP includes iron oxide nanoparticles (IONPs), which have been widely used for diagnostic imaging [136]. In this context, another trend is the incorporation of theranostics under clinical trials. They are multifunctional nanocompounds, which are well-designed by combining diagnostic and therapeutic capabilities into a single biocompatible and biodegradable nanoparticle [136,137]. This combination can offer synergistic advantages in comparison to standard imaging or treatment alone. They can facilitate the diagnosis and treatment of diseases at an early stage, provide interesting information on drug distribution to target sites, and enable the monitoring of the response to therapy. They are the most promising candidates for their application in a personalized disease [138].

To date, several theranostics are in clinical trials, and some of them are expected to be approved for clinical use in a very short time (Appendix A).

## 7. Future Perspectives of Nanopharmaceuticals Targeting TME

Active targeting has become the new trend in nanomedicine. It has reached a high level of precision and selectivity by exploiting molecules overexpressed just on the cancer cell surface, which facilitates the uptake of nanoparticles in tumor cells. In addition, new targets have emerged in recent years to become promising strategies for cancer therapy. Combining CSC marker targeting and drugs loaded onto nanomaterials can be effective against CSCs, increasing specificity and selectivity. Novel tools that aim to detect, characterize, and eliminate CSCs with enhanced efficacy are ongoing [139,140]. A considerable number of different nanoparticles have been evaluated, from anti-CSC ADCs [141], to nanoparticle-based RNAi [142], to nano-based thermotherapy [143].

Furthermore, accumulating evidence shows that the tumor microenvironment can re-program tumor initiation, growth, invasion, metastasis, and responses to therapies. Investigation and treatment have switched from a tumor cell model to a TME one, considering the increasing significance of TME in cancer biology. A promising strategy for TME targeting is the use of TAM-directed radiotracers and iron oxide nanoparticles for monitoring cancer immunotherapies with PET and MRI technologies. TAM-directed imaging probes can be designed to include immune-modulating properties, thereby leading to combined diagnostic and therapeutic (theranostic) effects [144].

## 8. Challenges in the Clinical Translation of Nanopharmaceuticals

Nanotechnology is very promising in the development of cancer therapies; however, it is important to address several issues for clinical use.

### 8.1. Costs, Production, and Toxicology Limitations

Economic issues are key challenges to guaranteeing the success of nanopharmaceuticals. The high costs of the raw materials and the production process make their products very expensive. For example, the manufacturing of drugs such as Abraxane^TM^ and Doxil^TM^ is far more expensive than the production of their free-drug counterparts (paclitaxel and doxorubicin) [145]. The entire process of commercializing a novel nanodrug is estimated to last for more than 10–15 years and cost around $1 billion [146]. Thus, the clinical benefits of nanomedicines should be clear to justify higher prices compared to conventional therapeutics [147]. The scalable and controlled manufacturing of nanomedicines under good manufacturing practices (GMP) conditions presents unique challenges since even small variations in the process can significantly affect properties such as size, shape, composition, drug loading and release, biocompatibility, toxicity, and the in vivo outcome [148,149,150]. Therefore, nanomedicine products should be characterized on a batch-to-batch basis, using multiple methods.

Moreover, nanomedicines intended for humans must deal with special issues such as their sterility [151]; thus, finding appropriate sterilization but not compromising the physicochemical properties and stability of the therapeutic molecules is one of the major challenges in nanomedicine development. Nanomedicines based on biological molecules such as proteins require special consideration due to their high susceptibility to degradation by sterilization techniques [152]. In this context, special consideration must be taken to the contamination by endotoxin. Endotoxin can lead to serious health issues, and more than 30% of nanoformulations fail in early preclinical development owing to endotoxin contamination [153]. Thus, the endotoxin level of nanomedicines must be carefully evaluated using appropriate methods [154]. The characterization of the stability and storage aspects (shelf-life) of nanomedicine products is also challenging [155]. Moreover, storage conditions in aqueous solutions or even in a lyophilized form can alter the nanomedicine properties [156].

The toxicological effects of nanomaterials must be evaluated, but some toxic effects are not yet completely known. Some derived toxicological data conflict and are inconsistent, except for a few observations reported so far. It is mandatory to implement guidelines to standardize preclinical nanomedicine research, which could promote reproducibility, quantitative comparisons, meta-analyses, and modeling, which would increase the applicability, cost, efficacy, and toxicity of nanoformulations and help to translate basic research into clinical practice in a very short time. In addition, regulatory issues are relevant for the development of technologies to characterize nanopharmaceuticals and monitor their quality. Regulatory decisions on nanomedicine therapeutics are based on the individual assessment of benefits and risks, a process that is time-consuming and may result in regulatory delays for nanomedicine products [157]. These actions should include integrating efforts from international consortia comprising academics, clinicians, pharmaceutical companies, and regulatory authorities to increase the clinical impact and patient performance of these antitumoral nanopharmaceuticals.

### 8.2. Clinical Translation of Nanopharmaceuticals

Numerous new papers evaluating a great variety of nanoformulations have been published lately, ranging from inorganic to organic nanoparticles, of several formulation and fabrication procedures and achieving high versatility, controllable size, and shape. They are functionalized for targeted therapy and are loaded with several chemotherapeutics and active molecules. However, good results in preclinical studies are in contrast with the low rate of clinical translation. In fact, a few nano-based drugs have been currently approved for use in clinics. Van der Meel and colleagues [158] proposed several strategic directions: firstly, the application of smart strategies for patient stratification in cancer nanomedicine, including probes and protocols to assess the tumor microenvironment, and imaging-based tumor accumulation, which may identify individuals suitable for inclusion in clinical trials, leading to refined clinical trials. Secondly, they proposed using smart strategies for modular (pro)drug and drug-delivery system design, as well as library screening, which will help to maximize the chances of formulations developed and tested preclinically, and, finally, they reported that the careful rational design of pharmacological combination regimens will amplify the pharmacokinetic and/or pharmacodynamic benefits. These smart strategies will improve nanomedicine performance, translation, and exploitation.

## 9. FDA and EMA Regulatory Guidelines

The FDA, in collaboration with the National Nanotechnology Institute (NNI) and the Nanotechnology Characterization Laboratory (NCL) in the U.S., has developed programs to coordinate efforts in nanoscale science, engineering, and technology. The FDA programs include draft guidance for the industry and a series of five “Final Guidances” related to nanotechnology products. These documents discuss the use of nanotechnology or nanomaterials in FDA-regulated products [159]. The FDA participates in many activities intended to implement a science-based approach to the regulation of products that involve nanomaterials or applications of nanotechnology, builds regulatory science knowledge, and facilitates collaborations and partnerships with stakeholders. It also facilitates innovation, helps ensure timely and clear communication regarding FDA’s nanotechnology activities, and safeguards the public’s health. Similarly, the EMA, in collaboration with the European Technology Platform on Nanomedicine (ETPN) and the European Nano-characterization Laboratory (EU-NCL), is also working to create regulatory guidelines for the evaluation of nanomedicine products. In 2011, the Committee for Medicinal Products for Human Use (CHMP) commissioned the EMA to develop a series of four reflection papers on current scientific and regulatory thinking for nanomedicines [160]. Both the FDA and EMA are members of the Innovation Task Force (ITF), which is an international and multidisciplinary group that includes scientific, regulatory, and legal competencies for nanotechnology products [161].

## Figures and Tables

**Figure 1 biomolecules-12-00784-f001:**
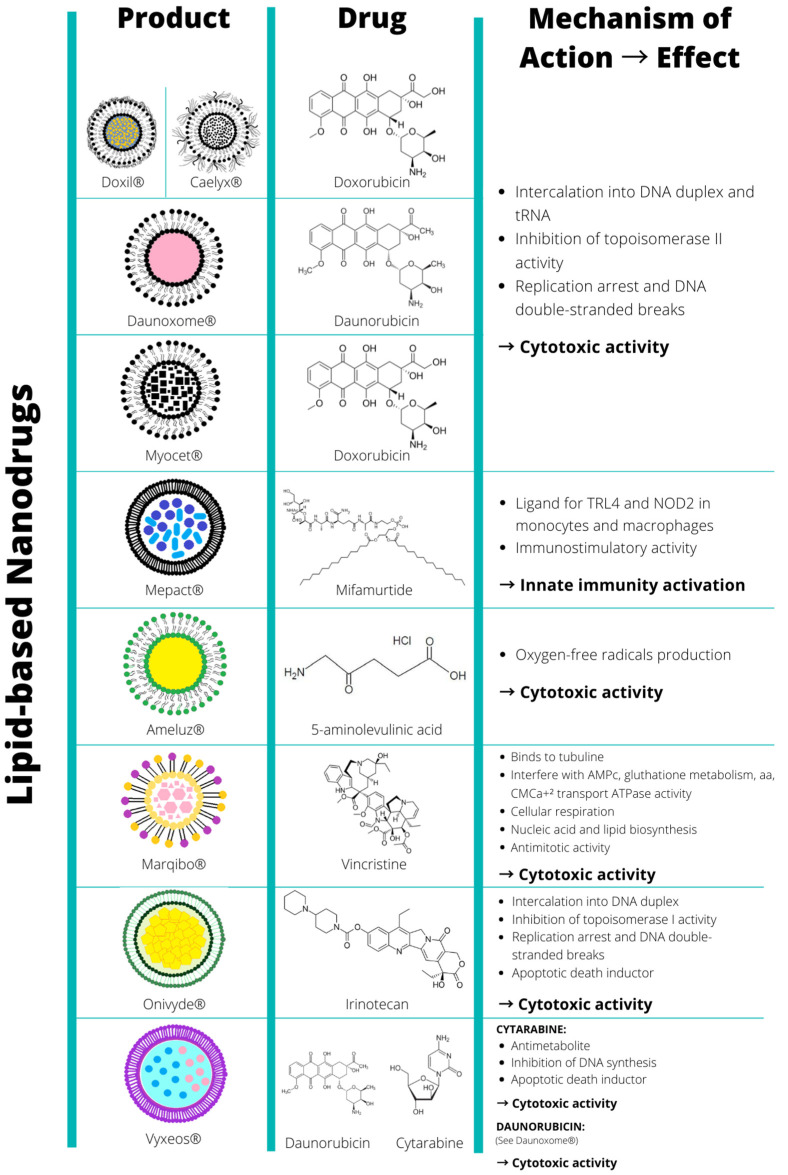
Lipid-based approved and marketed nanopharmaceuticals FDA and/or EMA-approved nanomedicines from 1995 to 2022 (last accessed May 2022), cataloged by their nature, encapsulated drug, their mechanism of action, and their induced effects.

**Figure 2 biomolecules-12-00784-f002:**
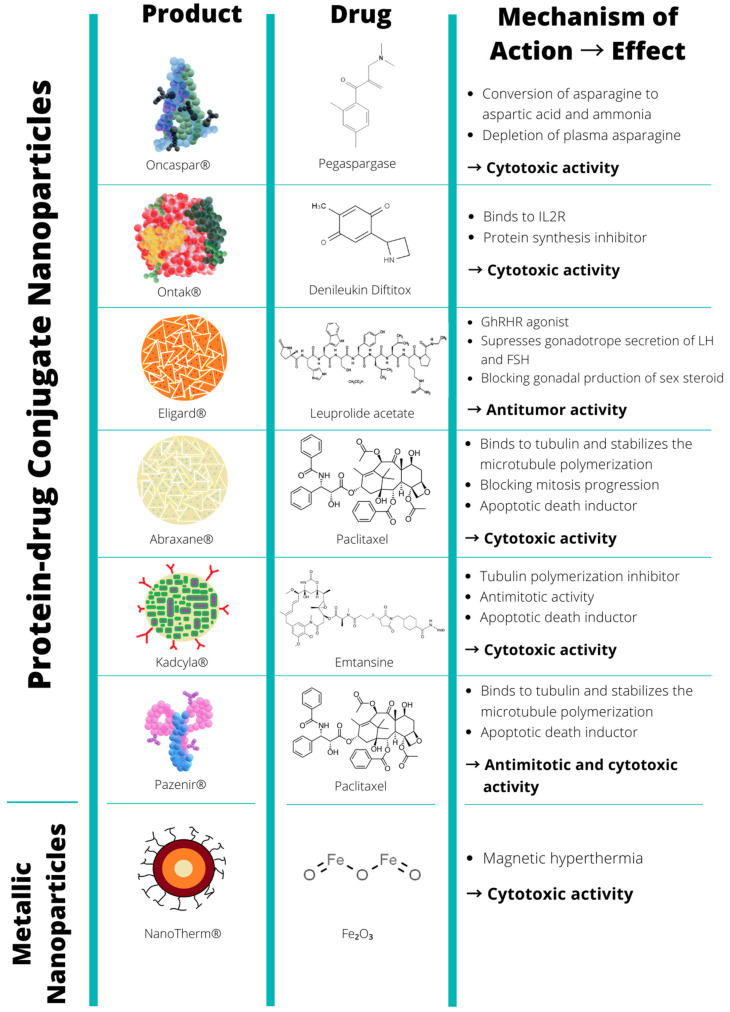
Protein- and metallic-based approved and marketed nanopharmaceuticals. FDA and/or EMA-approved nanomedicines from 1995 to 2022 (last accessed May 2022), cataloged by their nature, encapsulated drug, their mechanism of action, and their induced effects.

**Figure 3 biomolecules-12-00784-f003:**
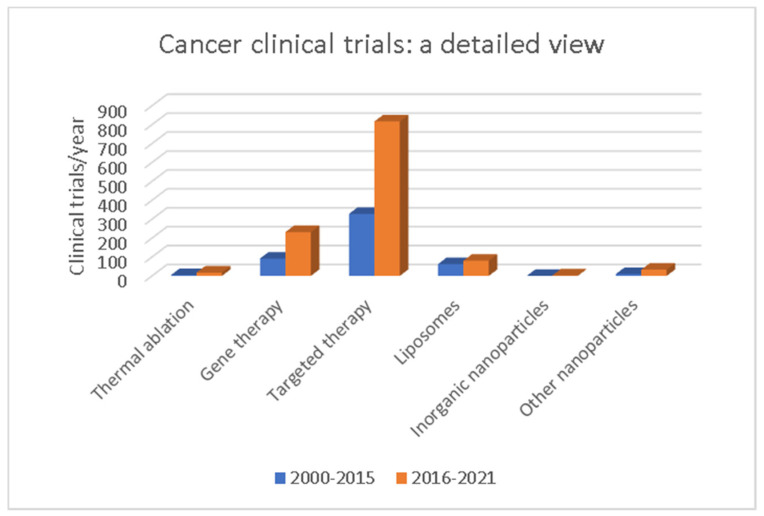
The status of clinical trials using nano–based formulations for cancer therapy. The number of the total and complete clinical trials, currently registered on www.clinicaltrials.gov (accessed on 28 April 2022), undergoing nano-based drugs treatment alone or in combination with other therapeutics during the periods 1995–2015 and 2015–2021. The results are expressed as the number of clinical trials divided by the number of years per period. In the last 10 years, second-generation DDS using active targeting have been the most widely tested systems in clinical trials.

**Table 1 biomolecules-12-00784-t001:** Approved and marketed nanopharmaceuticals for cancer therapy.

Nanostructure	Product^TM^	Nanotechnology Platform	Drug	Nanoformulation Advantages	TME Targeting	Indication	Company	Approval (Year)
**Lipid based nanoparticles**	Doxil	PEGylatyed STEALTH^®^ liposomes composed of MPEG-DSPE, HSPC, CHO.	Doxorubicin	↑ blood circulation time ↑ tumor uptake (EPR) ↓ cardiotoxicity	Cancer and stroma cells	Kaposi’s sarcoma, ovarian Ca, multiple myeloma	Ortho Biotech	FDA (1995)
	Caelyx	PEGylated liposomal doxorubicin composed of MPEG-DSPE, HSPC, CHO	Doxorubicin	↑ blood circulation time ↑ tumor uptake (EPR) ↓ cardiotoxicity	Cancer and stroma cells	Metastatic breast, Ca., ovarian Ca., Kaposi’s sarcoma, and multiple myeloma	Schering-Plough	EMA (1996)
	DaunoXome	Citrate salt of daunorubicin encapsulated in non-pegylated liposomes composed of DSPC and CHO (2:1 MR)	Daunorubicin	↓ protein binding ↑ blood circulation time ↑ tumor uptake (EPR) ↓ cardiotoxicity	Cancer and stroma cells	Kaposi’s sarcoma	Galen	FDA (1996)
	Myocet	Liposomal doxorubicin (non-PEGylated) composed of PC, CHO, citric acid, and NaOH	Doxorubicin	↑ blood circulation time ↑ tumor uptake (EPR) ↓ cardiotoxicity	Cancer and stroma cells	Metastatic breast Ca.	Teva UK	EMA (2000)
	Mepact	Liposomal mifamurtide (fully synthetic analogue of a component of Mycobacterium sp. cell wall) composed of POGP, DGPS, MS	Mifamurtide	↑ blood circulation time ↑ tumor uptake (EPR) ↓ toxicity	Macrophages	Osteosarcoma	Millenium	EMA (2009)
	Ameluz	Gel containing 5-aminolevulinic acid, E211, SoyPC, and PG	5-aminolevulinic acid	sustained release ↓ toxicity	Cancer and stroma cells	Superficial and/or nodular basal cell carcinoma	Biofrontera Bioscience GmbH	EMA (2011)
	Marqibo	Liposomal vincristine composed of SM and CHO	Vincristine	↑ blood circulation time ↑ tumor uptake (EPR) ↓ toxicity	Cancer and stroma cells	Acute lymphoid leukaemia	Spectrum	FDA (2012)
	Onivyde	Nanoliposomes composed of DSPC, CHO, MPEG-2000-DSPE	Irinotecan	↑ blood circulation time ↑ tumor uptake (EPR) ↓ toxicity	Cancer and stroma cells	Pancreatic Ca., Colorectal Ca.	Merrimack	FDA (2015)
	Vyxeos	Nanoliposomes composed of DSPC, DEPG, and CHO	Daunorubicin Cytarabine	↑ blood circulation time, ↑ accumulation in bone marrow	Cancer and stroma cells	Acute myeloid leukaemia	Jazz Pharmaceuticals	EMA (2018)
**Protein-drug conjugates**	Oncaspar	Covalent conjugate of L-asparaginase with mPEG, MSP, Na2HPO4, Heptahydrate, and NaCl	Pegaspargase	↑ blood circulation time ↑ tumor uptake (EPR)	Cancer cells	Acute lymphoblastic leukaemia	Les Laboratoires Servier	FDA (1994)
	Ontak	Recombinant cytotoxic protein composed of diphtheria toxin fragments A and B (Met1-Thr387)-His and human IL-2 (Ala1-Thr133)	Denileukin Diftitox	↑ blood circulation time ↑ tumor uptake (EPR) ↑ selectivity ↓ severe toxicity	Activated T-cells	Cutaneous T-cell lymphoma	Les laboratoires Servier	FDA (1999)
	Eligard	Polymeric matrix of leuprorelin acetate composed of PLGA (85:15) NMP and LA	Leuprorelin acetate	↑ blood circulation time ↑ tumor uptake (EPR)	Cancer cells	Prostate cancer	Recordati Industria Chimica e Farmaceutica	FDA (2002)
	Abraxane	Colloidal suspension without solvent of paclitaxel bound to albumin (active substance) in the form of a spherical nanoparticle	Paclitaxel	↑ Solubility ↑ blood circulation time ↑ tumor uptake (EPR) ↓ severe toxicity	Cancer and stroma cells	Breast Ca. Non-small lung Ca., Pancreatic Ca.	American Biosciencem, Inc.	FDA (2005)
	Kadcyla	Trastuzumab, covalently linked to DM1 via the stable thioether linker MCC	DM1 (or Emtansine)	↑ blood circulation time ↑ tumor uptake (EPR) ↑ selectivity ↓ toxicity	Cancer cells	HER2+ breast Ca.	Roche Genentech	EMA (2013) FDA (2013)
	Pazenir	Paclitaxel formulated as albumin bound nanoparticles. Powder for dispersion for infusion	Paclitaxel	↑ Solubility ↑ blood circulation time ↑ tumor uptake (EPR) ↓ severe toxicity	Cancer cells	Metastatic breast Ca., metastatic adenocarcinoma of the pancreas, non-small cell lung Ca.	Ratiopharm GmbH	EMA (2019)
**Metallic nanoparticles**	NanoTherm	Nanoparticles of superparamagnetic iron oxide coated with amino silane	Fe_2_O_3_	↑ blood circulation time ↑ tumor uptake (EPR) -heat production under stimulation with EMF -teranostic properties	Residual cancer and stroma cells	Glioblastoma, prostate, and pancreatic Ca.	Magforce	EMA (2013)

Abbreviations: ↑: increase; ↓: decrease; Ca.: cancer; MR: molar ratio; MPEG-DSPE: N-(carbonyl-methoxypolyethylene glycol 2000)-1,2-distearoyl-sn-glycero3-phosphoethanolamine sodium salt; HSPC: fully hydrogenated soy phosphatidylcholine; DGPS: 1,2-Dioleoyl-sn-glycero-3-phospho-L-serine; DSPC: distearoylphosphatidylcholine; POGP: 1-Palmitoyl-2-oleoyl-sn-glycero-3-phosphocholine; CHO: cholesterol; E211: Sodium benzoate; SoyPC: soybean phosphatidylcholine; PC: phosphatidylcholine; DSPC: 1,2-distearoyl-sn-glycero-3-phosphocholine; mPEG: Monomethoxypolyethylene glycol; MPEG-2000-DSPE: methoxy-terminated polyethylene glycol (MW 2000)-distearoylphosphatidyl ethanolamine; DEPG: Distearoylphosphatidylglycero; PG: propylene glycol; SM: sphingomyelin; MSP: sodium dihydrogen phosphate (monohydrate); MS: monosodium salt; Na2HPO4: disodium phosphate; NaCl: sodium chloride; NaOH: sodium hydroxide; PLGA: poly(lactic-co-glycolic acid); NMP: N-Methyl-2-Pyrrolidone; and LA: leuprolide acetate.

**Table 2 biomolecules-12-00784-t002:** The approved antibody drug conjugates for cancer therapy.

Drug	Product^TM^	Molecular Target	Cell Targeting	Company	Approval (Year)	Indication
Gemtuzumab ozogamicin	*Mylotarg*	CD33	Myeloid stem cells, myeloblasts, monoblasts, monocytes/macrophages, granulocyte precursors, and mast cells	Pfizer/Wyeth	2017; 2000	Relapsed acute myelogenous leukemia (AML)
Brentuximab vedotin	*Adcetris*	CD30	Lymphoid cells	Seattle Genetics, Millennium/ Takeda	2011	Relapsed HL and relapsed sALCL
Inotuzumab ozogamicin	*Besponsa*	CD22	B-cells	Pfizer/Wyeth	2017	Relapsed or refractory CD22-positive B-cell precursor acute lymphoblastic leukemia
Moxetumomab pasudotox	*Lumoxiti*	CD22	Leukemia cells	Astrazeneca	2018	Relapsed or refractory hairy cell leukemia (HCL)
Polatuzumab vedotin-piiq	*Polivy*	CD79	B-cells	Genentech, Roche	2019	Relapsed or refractory (R/R) diffuse large B-cell lymphoma (DLBCL)
Enfortumab vedotin	*Padcev*	Nectin-4	Cancer cells	Astellas/ Seattle Genetics	2019	Locally advanced or metastatic urothelial cancer who have received a PD-1 or PD-L1 inhibitor, and a Paclitaxel-containing therapy
Trastuzumab deruxtecan	*Enhertu*	HER2	Cancer cells	AstraZeneca/ Daiichi Sankyo	2019	Unresectable or metastatic HER2-positive breast cancer patients who have received two or more prior anti-HER2-based regimens
Sacituzumab govitecan	*Trodelvy*	Trop-2	Cancer cells	Immunomedics	2020	Metastatic triple-negative breast cancer (mTNBC) patients who have received at least two prior therapies (for patients with relapsed or refractory metastatic disease)
Belantamab mafodotin-blmf	*Blenrep*	BCMA	B-cells	GlaxoSmithKline	2020	Relapsed or refractory multiple myeloma
Loncastuximab tesirine-lpyl	*Zynlonta*	CD19	B cells and follicular dendritic cells	ADC Therapeutics	2021	Large B-cell lymphoma
Tisotumab vedotin-tftv	*Tivdak*	Tissue factor	Cancer and stroma cells of TME	Seagen Inc	2021	Recurrent or metastatic cervical cancer

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
