# Peer review of "Nano-Based Approved Pharmaceuticals for Cancer Treatment: Present and Future Challenges"

_biomolecules, 2022, doi:10.3390/biom12060784_

Round 1

Reviewer 1 Report

Comments:

Major comments:

In this manuscript, the authors first introduced cancer and the rationale for the development of a nano-based drug delivery system. Then, goes on to talk about targeting TME in cancer therapy followed by modes of targeting. After that, the authors elaborated on approved nanopharmaceuticals which is very informative. However, TME targeting, although it is important information for cancer therapy, is not adding much value to the overall article, it is somehow disconnected. None of the approved drugs are referenced back to what it is specifically targeting, cancer TME or CSCs, etc. I suggest adding information on approved drugs on their specific targets.

CSCs again was mentioned in the future and perspective as a major target for cancer therapy. Please include current studies (either under the CSCs targeting or here). I believe active targeting has high potential but still is challenging leading to only a small percentage of drugs reaching the target site. Therefore, it is still sub-optimal. Please include theragnostic particles in the future and perspective.

Line 70:” In this review, we will provide an overview of the nanotechnological advantages applied to cancer therapy and the rationale of the approved and marketed nanopharmaceuticals. Moreover, we highlight the new approaches that are currently under clinical trials and discuss their advantages, limitations, and future perspectives such as their integration within the burgeoning personalized medicine.” This is either missing or difficult to point out. Please revise.

Minor Comments:

1.       Through the manuscripts, some sentences are either super long or confusing. Please revise some of the examples including Line 55, 61, 94, 161, 258, and 299. Please break the sentences for clarification.

2.       “Targeting cancer stem cells in TME” subsection talks about CSCs and their contribution to cancer initiation and progression. The CSCs markers are dependent on tumor types which may be the limiting factor in designing nano-pharmaceuticals for targeted cancer therapy. Please provide design considerations and the challenges involved in targeting CSCs for cancer treatment. Also, provide examples of nano-pharmaceuticals that are currently under preclinical or clinical trials.

3.       Line 227, please provide examples where they used stromal targeting drugs in combination with cancer-targeting agents, delivered together or in sequences.

4.       Line 234: add a reference

5.       Line 254: Please elaborate on metastasis initiating cells

6.       Line 286: Multidrug resistance is primarily due to the high systemic dosage used to achieve a sufficient amount of drug at the diseased site to trigger a therapeutic response.

7.       Please include challenges or limitations for lipid-based, protein-based, and metallic-based nanopharmaceuticals.

8.       Please include any combination therapies for cancer that is currently in preclinical or clinical trials.

9.       Line 268, active targeting facilitated the cellular uptake but “guaranteeing” is a strong word. Please revise.

10.   Line 636 is repetitive.

Author Response

May 31, 2022

we really appreciate and have carefully taken into consideration the suggestions, comments, and questions raised about our manuscript, which will undoubtedly help to improve the paper.

  Find below the detailed answers point per point.

Answers to Reviewer 1

In this manuscript, the authors first introduced cancer and the rationale for the development of a nano-based drug delivery system. Then, goes on to talk about targeting TME in cancer therapy followed by modes of targeting. After that, the authors elaborated on approved nanopharmaceuticals which is very informative. However, TME targeting, although it is important information for cancer therapy, is not adding much value to the overall article, it is somehow disconnected. None of the approved drugs are referenced back to what it is specifically targeting, cancer TME or CSCs, etc. I suggest adding information on approved drugs on their specific targets.

Thank you for the suggestion. In this revised manuscript, we created a new column in Tables 1 and 2 identified as ”Targeting TME” and “Cell targeting”, respectively. There, we included information on approved drugs and their specific cellular targets. In addition, the relevance of targeting the TME is also described in section 7.

CSCs again were mentioned in the future and perspective as a major target for cancer therapy. Please include current studies (either under the CSCs targeting or here). I believe active targeting has high potential but still is challenging leading to only a small percentage of drugs reaching the target site. Therefore, it is still sub-optimal. Please include theragnostic particles in the future and perspective.

As reviewer´s suggestions, in section 7 (Future perspectives of nanopharmaceuticals targeting TME)  we included information about the current strategies and studies involving nanoparticles that led to eliminating CSCs.  As consequence, some new references have been added.

On the other hand, theragnostic nanoparticles were described in section 6, however, we missed to include them in the future and perspectives section 7. Following the reviewer´s suggestions, we have introduced them in this section, and we have also included five clinical trials in Supplementary Table 1.

Line 70:” In this review, we will provide an overview of the nanotechnological advantages applied to cancer therapy and the rationale of the approved and marketed nanopharmaceuticals. Moreover, we highlight the new approaches that are currently under clinical trials and discuss their advantages, limitations, and future perspectives such as their integration within the burgeoning personalized medicine.” This is either missing or difficult to point out. Please revise.

We rewrote the paragraph for clarification. We hope it will be comprehensive in its present form.

Minor Comments:

  1. Through the manuscripts, some sentences are either super long or confusing. Please revise some of the examples including Line 55, 61, 94, 161, 258, and 299. Please break the sentences for clarification.

Thank you for the comment. We rewrote the sentences for clarification.

  1. “Targeting cancer stem cells in TME” subsection talks about CSCs and their contribution to cancer initiation and progression. The CSCs markers are dependent on tumor types which may be the limiting factor in designing nano-pharmaceuticals for targeted cancer therapy. Please provide design considerations and the challenges involved in targeting CSCs for cancer treatment. Also, provide examples of nano-pharmaceuticals that are currently under preclinical or clinical trials.

The information is now included in section 6 “future perspectives of nanopharmaceuticals targeting TME”. We provided some examples of promising strategies and nanopharmaceuticals used for CSCs therapy that is currently under experimental and /or clinical trials. We also added some references.

  1. Line 227, please provide examples where they used stromal targeting drugs in combination with cancer-targeting agents, delivered together or in sequences.

Nab-paclitaxel is the most used stromal and anti-tumor agent in combination with other conventional therapies. It is present in about 60% of ongoing clinical trials. As the reviewer suggested, we added this information to the text. In addition, Supplementary Table 1 shows, in detail, the different clinical trials using nab-paclitaxel and other nanoparticles in monotherapy or combined with other conventional cancer treatments.

  1. Line 234: add a reference

A reference is added. Thank you.

  1. Line 254: Please elaborate on metastasis initiating cells

                Thank you for the suggestion. We elaborated on the metastasis initiating cells (MICs)  more in detail in section 3.3.

  1. Line 286: Multidrug resistance is primarily due to the high systemic dosage used to achieve a sufficient amount of drug at the diseased site to trigger a therapeutic response.

Thank you. We included this information in the text.

  1. Please include challenges or limitations for lipid-based, protein-based, and metallic-based nanopharmaceuticals.

As the reviewer suggested, challenges and limitations for nanoparticle-based therapies have been described in detail in a new sub-section within section 8 (“Challenges in the clinical translation of nanopharmaceuticals). We exposed the main limitations of using nanoparticles for cancer therapy which are the costs, production methods, and toxicology. They limit the clinical translation of these therapies.

  1. Please include any combination therapies for cancer that is currently in preclinical or clinical trials.

Supplementary table 1 shows, in detail, the different clinical trials using nanoparticles in monotherapy or combined with other conventional cancer treatments. Nab-paclitaxel formulation is the most used nanoparticle in combination, being present in up to 60% of ongoing clinical trials. We added this information to the text. 

  1. Line 268, active targeting facilitated the cellular uptake but “guaranteeing” is a strong word. Please revise.

We revised the manuscript, and we changed the word “guaranteeing” to “facilitate”. Thank you for the suggestion.

  1. Line 636 is repetitive.

We deleted the repetitive information and rewrote the text for clarification. Thank you.

Sincerely,

Maria Virtudes Céspedes, PhD

Gynecologic & Peritoneal Oncology Group

Reviewer 2 Report

I have read the review, 'Nano-based approved pharmaceuticals for cancer treatment: present and future challenges' with high interests. This review can be published after major revisions.

1. The table needs to include more examples of inorganic nano-based formulations. These are completely ignored it seems: Inorganic nanoparticles in clinical trials and translations - ScienceDirect

2. Please include a section about nanoparticles regulatory guidance from agencies like FDA: Nanotechnology Guidance Documents | FDA

3.  What are the major limitations of the nanomedicines? Please discuss more in terms of toxicity, pharmacokinetics, targeting, degradability, cost and production challenges. Please provide few examples for each section. Following paper can be consulted for writing these sections. 

4. The figures resolution need to be better for clarity.

5. Some recent ref can be added: Theranostics, 2019, 9 (25), 7730-7748

Author Response

May 31, 2022

we really appreciate and have carefully taken into consideration the suggestions, comments, and questions raised about our manuscript, which will undoubtedly help to improve the paper.

Find below the detailed answers point per point.

 Answers to Reviewer 2

I have read the review, 'Nano-based approved pharmaceuticals for cancer treatment: present and future challenges' with high interests. This review can be published after major revisions.

1. The table needs to include more examples of inorganic nano-based formulations. These are completely ignored it seems: Inorganic nanoparticles in clinical trials and translations – ScienceDirect

Our main purpose has been to compile the information about the used nanoparticles only for cancer therapy. Following the reviewer´s suggestion, the inorganic nano-based formulations for this purpose have been added in section 6 and figure 3. In addition, five ongoing clinical trials with inorganic-based nanoparticles have been included in Supplementary Table 1.

Please include a section about nanoparticles regulatory guidance from agencies like FDA: Nanotechnology Guidance Documents | FDA

We created a new section 9 entitled “FDA and EMA regulatory guidelines” to introduce this information. Thank you very much for the suggestion.

What are the major limitations of the nanomedicines? Please discuss more in terms of toxicity, pharmacokinetics, targeting, degradability, cost and production challenges. Please provide few examples for each section. Following paper can be consulted for writing these sections. 

Thank you very much for the comments. We included these topics in section 8, entitle “Challenges in the clinical translation of nanopharmaceuticals”. We created two sub-sections:  the 8.1 Costs, production, and toxicology limitations and the 8.2 Clinical translation of nanopharmaceuticals. We think they will improve the article. We exposed the main limitations of using nanoparticles for cancer therapy which are the costs, production methods, and toxicology. They limit the clinical translation of these therapies. We added some new references.

In addition, we described some strategies reported by Van der Meel and cols [158] to  improve nanomedicine performance, translation and exploitation.

The figures resolution needs to be better for clarity.

Done. We improved the final resolution of the figures. We hope they will be comprehensive in their present form.

  1. Some recent ref can be added: Theranostics, 2019, 9 (25), 7730-7748

We included this reference in the section  (“Future perspectives of nanopharmaceuticals targeting TME). Thank you for the suggestion.

Sincerely,

Maria Virtudes Céspedes, Ph D

Gynecologic & Peritoneal Oncology Group

Round 2

Reviewer 1 Report

Thank you for addressing all the comments. The manuscript is very informative and would be significant contribution in the field.

Reviewer 2 Report

All the reviewer's comments were being taken care of by the authors and hence the current version of the manuscript can be accepted in its present form.